# Triple-Negative Breast Cancer with High Levels of Annexin A1 Expression Is Associated with Mast Cell Infiltration, Inflammation, and Angiogenesis

**DOI:** 10.3390/ijms20174197

**Published:** 2019-08-27

**Authors:** Maiko Okano, Masanori Oshi, Ali Linsk Butash, Eriko Katsuta, Kazunoshin Tachibana, Katsuharu Saito, Hirokazu Okayama, Xuan Peng, Li Yan, Koji Kono, Toru Ohtake, Kazuaki Takabe

**Affiliations:** 1Breast Surgery, Department of Surgical Oncology, Roswell Park Comprehensive Cancer Center, Buffalo, NY 14263, USA; 2Department of Breast Surgery, Fukushima Medical University School of Medicine, Fukushima 960-1295, Japan; 3Department of Gastrointestinal Tract Surgery, Fukushima Medical University School of Medicine, Fukushima 960-1295, Japan; 4Department of Biostatistics & Bioinformatics, Roswell Park Comprehensive Cancer Center, Buffalo, NY 14263, USA; 5Department of Surgery, University at Buffalo Jacobs School of Medicine and Biomedical Sciences, The State University of New York, Buffalo, NY 14203-1121, USA; 6Department of Breast Surgery and Oncology, Tokyo Medical University, Tokyo 160-8402, Japan; 7Department of Surgery, Yokohama City University, Yokohama 236-0004, Japan; 8Department of Surgery, Niigata University Graduate School of Medical and Dental Sciences, Niigata 951-8510, Japan

**Keywords:** breast cancer, ANXA1, mast cell, inflammation, TNBC, CIBERSORT, GSEA

## Abstract

Annexin A1 (ANXA1) is a phospholipid-linked protein involved in inflammation, immune response, and mast cell reactivity. Recently, we reported that ANXA1 is associated with aggressive features of triple-negative breast cancer (TNBC); however, its clinical relevance remains controversial. We hypothesized that human TNBC with high expression of ANXA1 mRNA is associated with pro-cancerous immune cell infiltration, including mast cells, and with an aggressive phenotype. Clinical and RNA-seq data were obtained from The Cancer Genome Atlas (TCGA, *n* = 1079) and Molecular Taxonomy of Breast Cancer International Consortium (METABRIC) (*n* = 1904). TNBC patients had significantly higher levels of ANXA1 expression compared to the other subtypes in both TCGA and METABRIC cohorts (*p* < 0.001). ANXA1 protein expression was assessed by immunohistochemistry in Japanese TNBC patient cohort (*n* = 48), where 17 cases (35.4%) had positive ANXA1 staining, and their overall survival was significantly shorter compared with negative staining group (*p* = 0.008). The CIBERSORT algorithm was used to calculate immune cell infiltrations. ANXA1 high tumors were associated with activated mast cells and M2 macrophages (*p* > 0.01), but did not show any association with tumor heterogeneity nor cytolytic activity. High expression of ANXA1 group enriched inflammation, epithelial-to-mesenchymal transition (EMT), and angiogenesis-related genes in a gene set enrichment assay in both cohorts. To our knowledge, this is the first study to demonstrate that ANXA1 is associated with infiltration of mast cells and inflammation that is associated with the aggressive phenotype of TNBC, such as EMT and angiogenesis.

## 1. Introduction

Triple-negative breast cancer (TNBC), defined as a subtype expressing less estrogen receptor (ER), progesterone receptor (PgR), and human epithelial growth factor receptor (HER2) by immunohistochemistry, accounts for approximately 15% of all breast cancer. TNBC is a biologically and clinically highly aggressive subtype with poor survival [1,2,3] This is partly due to a lack of subtype-specific therapeutic agents, such as hormone therapy for ER-positive and Trastuzumab for HER2 positive subtypes, respectively. Currently, conventional chemotherapeutic agents are the mainstay of systemic therapy for TNBC and novel therapeutic targets are needed to improve survival. 

Annexin A1 (ANXA1) is a 37-kDa calcium-dependent phospholipid-linked protein that is known to play a role in inflammation and immune response [4,5]. However, the relationship between ANXA1 levels and patient survival in breast cancer has been inconsistent in prior studies [6,7]. We have previously reported that high expression of ANXA1 protein in gastric and colon cancer is associated with worse prognosis [8,9]. We further found that ANXA1 protein was associated with tumor invasiveness in 35 Japanese TNBC patients [10]. However, the limitation of this study was that the cohort size was small. Recently, another group reported that ANXA1 is associated with poor survival in TNBC, in a study that combined 23 cohorts of microarray gene expression data [11]. The limitation of that study was that they combined the datasets that were derived from different platforms and the authors did not utilize systemic quality control or statistical modeling to control batch effects related to the different platforms, genotypes, and phenotypes included in these experiments. In particular, the confounding factors of the batches and their gene-level classification were not addressed. Without well-developed design and statistical analysis, such confounding factors, it could have significant effects on the validity of the results, as pointed out previously [12,13,14]. Bhardwaj et al. reported that ANXA1 mRNA expression was associated with significantly shorter overall survival in basal-like breast cancer (BLBC) patients utilizing The Cancer Genome Atlas (TCGA) (Nature 2012) cohort that included only 890 breast cancer patients [15]. The latest TCGA cohort has increased the number of breast cancer patients by approximately 20%, now approaching 1100. Additionally, basal-like breast cancer is a subtype defined by gene expression profiles and is not practically useful in a clinical setting. 

ANXA1 has been reported to promote distant metastasis by inducing epithelial-to-mesenchymal transition (EMT) via TGF-β signals in an in vitro model [16]. ANXA1 was also reported to promote NF-κB activity and to upregulate inflammation using human cell lines and xenograft models [4,17].

Recently, the relationship between ANXA1 and mast cells has been highlighted. Mast cells are known to play a major role in allergic reactions by degranulation and release of mediators, including histamine [18], but its role in breast cancer remains controversial. Some reported that infiltration of mast cells in the tumor was associated with better patient survival [19], and the others reported that it was associated with high-grade tumors [20]. Several studies demonstrated the relationship between mast cells and tumor angiogenesis [21]. Ample studies have demonstrated that various substances are released by mast cells, such as VEGF and TGF-β, promote angiogenesis [22]. ANXA1 was also shown to regulate mast cell reactivity [23], and its release from mast cells acts on neutrophils to help promote an anti-inflammatory effect [24]. Our group has been studying the strong link between inflammation and cancer and has conducted several investigations [25,26,27]. Many researchers have previously reported that mast cells are involved in inflammation and the development and progression of cancer [28,29].

Given this background, we hypothesized that ANXA1 high tumors are involved in increased mast cell infiltration, increased inflammation, and angiogenesis-related gene expression, and finally, are associated with worse survival in TNBC.

## 2. Results

### 2.1. TNBC Patients with Positive ANXA1 Protein Stain Had Significantly Worse Overall Survival

We have previously demonstrated that TNBC cases with high ANXA1 protein expression were associated with aggressive features of breast cancer [10]. The limitation of that study was that all of the patients were treated before 2005 with a small cohort (*n* = 35). We updated the cohort by adding 13 cases from the period 2005–2011 and re-analyzed the whole cohort. ANXA1 protein expression in breast cancer tissues was calculated by using anti-ANXA1 antibody staining (Figure 1A,B). Within the tumors, the ANXA1 expression was mainly detected in the cytoplasm of cancer cells, stromal cells, and myoepithelial cells. Among 48 TNBC patients, 17 cases had positive ANXA1 staining. Overall survival (OS) was significantly shorter in patients with ANXA1 stain positive tumors compared with ANXA1 stain negative tumors (Figure 1). ANXA1 positivity in cancer cells had no statistical relationship with the clinical characteristics of the patients (Appendix A). 

### 2.2. Expression of ANXA1 mRNA Was Significantly Higher in TNBC than in Other Breast Cancer Subtypes

As shown in Figure 2, we found that TNBC patients had significantly higher levels of ANXA1 mRNA expression compared to the other subtypes of breast cancer patients in both cohorts; TCGA, which is mainly United States patients, and Molecular Taxonomy of Breast Cancer International Consortium (METABRIC), which is mainly United Kingdom and Canada patients (*p* < 0.001, respectively). One hundred and two Caucasians, 55 African Americans, and 5 Asians were included in the TNBC group in a TCGA (provisional) cohort, and there was no difference among the ethnicities (Appendix A). This result suggests that elevation of ANXA1 expression is not unique to TNBC in a Japanese population [10], but also applicable to United States and European populations.

### 2.3. ANXA1 Expression Was not Associated with Tumor Heterogeneity or Tumor Cytolytic Activity

Current dogma is that cancer with high mutation load and/or heterogeneity attracts immune cells, and those cells attack cancer cells using cytolytic activity. Thus, it was of interest whether ANXA1 expression associates with tumor heterogeneity, which was assessed by mutation load and a mutant allele tumor heterogeneity (MATH) algorithm, and with cytolytic activity score (CYT). There was no significant association between ANXA1 expression and mutation load, tumor heterogeneity, or cytolytic activity in either TCGA or METABRIC cohort (Figure 3). 

### 2.4. ANXA1 High Tumors Were Associated with Activated Mast Cells and M2 Macrophages, but Not with Neutrophils or Lymphocytes

Given the previous reports that demonstrated the role of ANXA1 in the attraction of several immune cells to cancer in animal models, we investigated whether this was the case in human primary TNBC utilizing the CIBERSORT algorithm in TCGA and METABRIC cohorts. We found that high ANXA1 tumors were significantly associated with high infiltration of activated mast cells (*p* < 0.01) consistently in both TCGA and METABRIC cohorts (Figure 4). The association between high ANXA1 expression and M2 macrophages was found only in a METABRIC cohort. On the other hand, we did not see any association of neutrophils, CD4+, or CD8+ T-cell infiltration with ANXA1 expression in either cohort, which is in agreement with our finding that ANXA1 expression was not associated with cytolytic activity. 

### 2.5. High Expression of ANXA1 Enriched Inflammation, Epithelial-to-Mesenchymal Transition, and Angiogenesis Gene Sets in GSEA in Both Cohorts

To investigate the mechanism of ANXA1 high expression tumors in TNBC patients, gene set enrichment analysis (GSEA) was conducted between ANXA1 high and low expression patient groups. GSEA demonstrated that approximately 10 of the 50 hallmark gene sets were significantly enriched in ANXA1 high expression tumors in TNBC with normalized *p* < 0.05 in TCGA cohorts. Many inflammation-related gene sets, such as TNF-α signaling, hypoxia, complement and interleukin signaling, as well as apoptosis gene sets, were enriched with ANXA1 high tumors, which were in agreement with the previous reports using in vitro system (Figure 5, and Appendix A). Epithelial-to-mesenchymal transition (EMT) and TGF-β signaling gene sets were also enriched with high ANXA1 expression, which is in agreement with a previous in vitro study [16]. Interestingly, angiogenesis gene sets were also enriched in high ANXA1 tumors, which is consistent with our cell composition results showing that those tumors have high mast cell infiltration. Strikingly, these results were almost completely mirrored in METABRIC only with higher significance (lower *p*-value), which is most likely because the number of patients in METABRIC is close to double that of TCGA. On the other hand, there were no significantly enriched gene sets in ANXA1 low expression tumors. 

## 3. Discussion

ANXA1 is known to be related to inflammatory pathways [30], cell proliferation, and the regulation of cell death signaling [31]. However, there have been controversies in its role in breast cancer and survival. To our knowledge, this is the first study to investigate the association between ANXA1 expression and cancer using two large independent cohorts. We found that ANXA1 high tumors enriched inflammation and EMT gene sets, which suggest that previously reported mechanisms using in vitro and in vivo experiments are in place in human cancer and are clinically relevant.

Strikingly, we found that ANXA1 high tumors significantly enriched angiogenesis gene sets were associated with high infiltration of activated mast cells. The role of tumor-infiltrating mast cells has been drawing attention in recent years. However, their impact on patients’ prognosis has been controversial [32,33]. It was reported that the density of mast cells increases in breast tumors after neoadjuvant chemotherapy, suggesting a strong association with tumor immunity [34]. Taken together, we cannot help but speculate that ANXA1 expression may play a role in mast cell-induced inflammation and angiogenesis in breast cancer, which is a different mechanism compared to that seen in other cancers, such as colon cancer, and that may be one of the reasons why clinical trials using anti-angiogenesis therapy have failed in breast cancer. 

ANXA1 continues to attract attention, and some reported that the expression of ANXA1 may be associated with chemotherapy response in TNBC [35]. Other authors reported that ANXA1 plays an important role in the tumor microenvironment [36].

Our study has several limitations. The first and biggest was that while our research has demonstrated the association between ANXA1 expression and mast cell infiltration, we have no data to suggest causality. The second was that TCGA and METABRIC databases have substantial missing values, especially information regarding metastasis. In addition, immune cells were assessed with a computational algorithm alone and not with flow cytometry or immunohistochemical staining (IHC) in this study. It would have been ideal to validate the association of ANXA1 expression and mast cell infiltration with protein expression as was previously done in survival analyses; however, no sample was available for flow cytometry that would allow such analyses. In the future, we hope to conduct a prospective study with a larger sample volume to overcome these issues.

## 4. Materials and Methods

### 4.1. Patient Cohorts

The patient clinical data and gene expression data (mRNA expression z-score of RNA-seq) were downloaded through cBioPortal from the latest The Cancer Genome Atlas (TCGA) cohort (TCGA provisional) as described previously [37,38,39,40,41]. Among 1081 female breast cancer patients with mRNA expression from RNA sequence data, survival data were available in 1079 patients. Level 3 Z-score normalized gene expression data, as well as overall survival data from 1904 patients in a Molecular Taxonomy of Breast Cancer International Consortium (METABRIC) cohort [42], were also downloaded from cBioPortal. Patients were classified as either high or low ANXA1 expression using the median of their mRNA expression as the cut-off [11,15]. Disease-free survival (DFS) was defined from the time of completion of primary treatment until clinical confirmation of tumor recurrence. Overall survival (OS) was defined using time of death. For disease-specific survival (DSS) analyses, the patients who died of other causes were excluded. CYT is the immune cytolytic activity score. It has previously been defined by calculating the geometric mean expression values of granzyme A (GZMA) and perforin (PRF1) [43]. These proteins are pivotal cytolytic effector molecules. MATH is the mutant-allele tumor heterogeneity level. This is derived by calculating the median of its mutant-allele fractions at tumor-specific mutated loci, which has been described in detail [44,45,46].

### 4.2. CIBERSORT

CIBERSORT, a bioinformatic algorithm that allows calculation of immune cell composition from gene expression profiles, was invented to estimate tumor-infiltrating immune cells in tumors [47] and was applied as described previously [48]. Immune cell fraction data was downloaded through The Cancer Imaging Archive (TCIA) (https://tcia.at/home) [49].

### 4.3. Gene Set Enrichment Analysis

GSEA was performed on TCGA and METABRIC cohorts using software provided by the Broad Institute (http://software.broadinstitute.org/gsea/index.jsp), as described previously [50,51]. We classified the patients into two groups according to ANXA1 expression using the mean of their gene expression range.

### 4.4. Clinical Samples for Protein Expression

All cases analyzed with immunohistochemistry underwent surgical resection at Fukushima Medical University between 2002 and 2011. The present study was approved by the Institutional Review Board and the Ethics Committee of Fukushima Medical University.

### 4.5. Immunohistochemical Staining and Evaluation

The immunohistochemical staining approach involved immunostaining formalin-fixed paraffin-embedded (FFPE) sections for ANXA1. The staining intensity was analyzed using previously reported methods [10]. These methods involve taking breast cancer specimens and fixing them in 10% formalin and embedding them in paraffin. The fixed specimens are then cut into 4 μm sections and stained with anti-ANXA1 antibody (clone 29; BD Biosciences, San Jose, CA, USA). ANXA1 expression was defined as high when more than 5% of cells per high-power field stained positive by immunohistochemistry, following the criteria used in our previous report [10]. Less than 5% staining of ANXA1 was defined as low expression.

### 4.6. Statistical Analysis

The differences in OS and DSS between ANXA high and low groups were analyzed using a Kaplan–Meier method with the log–rank test as described previously [51,52,53]. The clinicopathological characteristics were compared statistically using the chi-square test or the Fisher exact test. The differences between continuous values were compared using the Student’s *t*-test. A *p* < 0.05 was considered statistically significant. All statistical analyses were performed using Microsoft Excel 2010, R software (http:///www.rproject.org/) and Bioconductor (http://bioconductor.org/).

## 5. Conclusions

These results prove that high expression of ANXA1 is significantly associated with mast cell infiltration in TNBC, which can explain the strong association with angiogenesis. Our report reinforces the fact that breast cancer and inflammation show a strong relationship, which can add a new dimension to the future research and treatment of TNBC.

## Figures and Tables

**Figure 1 ijms-20-04197-f001:**
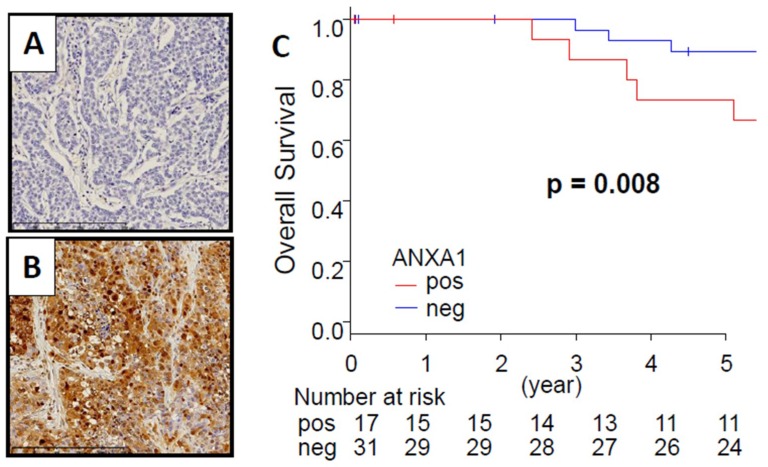
Annexin A1 (ANXA1) protein expression is associated with overall survival in a Japanese triple-negative breast cancer (TNBC) cohort. Representative images of ANXA1 immunohistochemical staining in breast cancer. (**A**,**B**) Negative and positive ANXA1 staining in breast cancer tissue. ×400 magnification. (**C**) Overall survival in a Japanese TNBC cohort. Positive and negative ANXA1 protein staining results are represented by the red and blue lines, respectively. Bold font indicates a significant difference. (*p* < 0.01).

**Figure 2 ijms-20-04197-f002:**
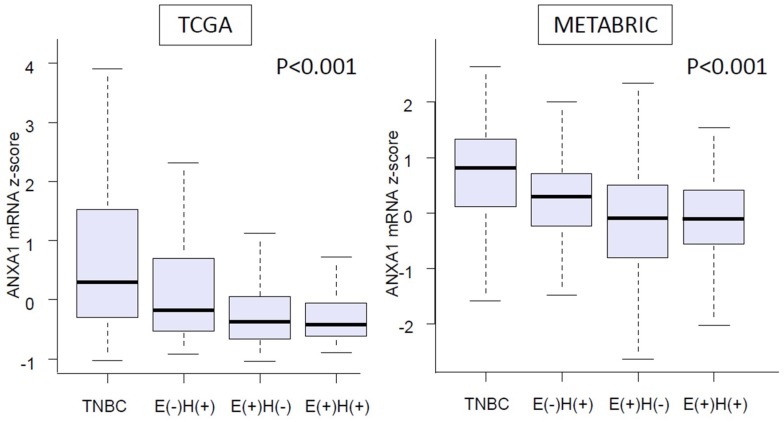
Expression level of ANXA1 mRNA in each subtype of breast cancer samples from a The Cancer Genome Atlas (TCGA) cohort (*n* = 1060) and a Molecular Taxonomy of Breast Cancer International Consortium (METABRIC) cohort (*n* = 1866). E = estrogen receptor, H = Her2 receptor.

**Figure 3 ijms-20-04197-f003:**
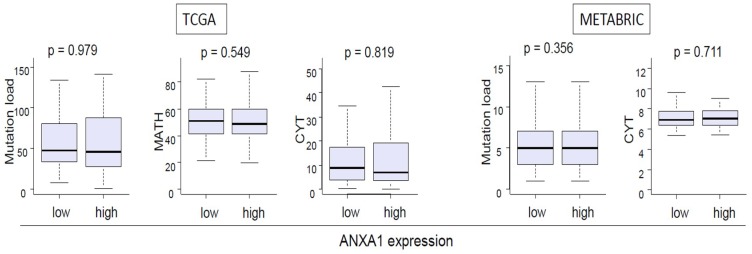
There is no significant difference in mutation load, mutant-allele tumor heterogeneity (MATH) score and Cytolytic Activity Score (CYT) between ANXA1 high and low TNBC in TCGA and METABRIC cohorts.

**Figure 4 ijms-20-04197-f004:**
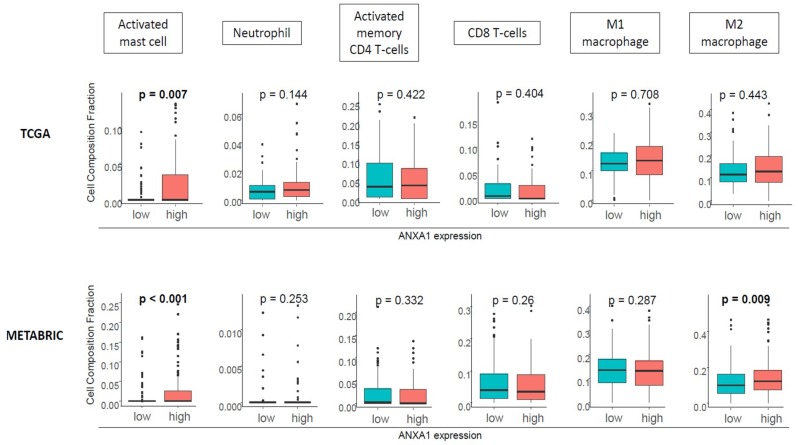
ANXA1 high tumors are associated with high infiltration of mast cells. CIBERSORT analysis demonstrated high infiltration of mast cells (*p* < 0.01) but not in the other cells in TCGA. Mast cells and M2 macrophages are significantly infiltrated in METABRIC (*p* < 0.01).

**Figure 5 ijms-20-04197-f005:**
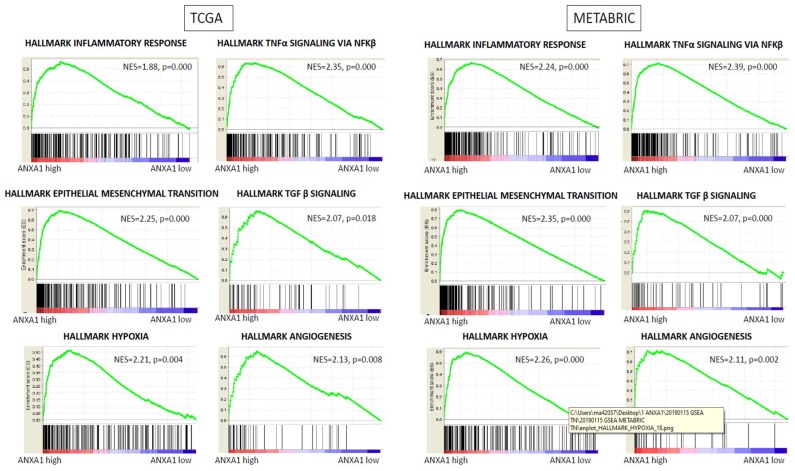
Gene set enrichment analysis (GSEA) showed that high ANXA1 expression tumors significantly enriched inflammatory response, TNF-α signaling via NFκB, epithelial-mesenchymal transition, TGF-β signaling, hypoxia, and angiogenesis gene sets in TNBC in both TCGA and METABRIC cohorts.

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
