# Peer review of "Triple-Negative Breast Cancer with High Levels of Annexin A1 Expression Is Associated with Mast Cell Infiltration, Inflammation, and Angiogenesis"

_ijms, 2019, doi:10.3390/ijms20174197_

Round 1

Reviewer 1 Report

The current study addressed the role, expression, and mechanisms/ biological effects of ANXA1 in triple negative breast cancers (TNBC).

 The paper is well written and comprehensive.

 There are only few minor points to address.

line 91-92: unclear sentence: “…enriched inflammation and angiogenesis related gene sets,”…Do you mean increased expression/activation of .. gene sets? It is necessary to re-construct this sentence and make it easier for understanding.

2.Figure 1: please specify in more details what does it mean:” High and low expression are represented by the red and blue lines” as in the figure legend it is marked as pos and neg. Does neg means low? And pos means high expression of ANXA? Correct if wrong.

The Fig 1 A and B shows only negative and positive staining. However, the authors wrote about low and high expression levels of ANXA1. It is necessary to present how the Low and High expressing tissues look like. The Immunohistochemistry should be extended in fig 1. Sub-titles should be improved. For instance: “ANXA1 high tumors were associated with activated mast cells…” should be “ANXA1 high tumors were associated with localization/presence of activated mast…etc”. The staining/immunohistochemistry for mast cells presence will add a value to this paper. If it is possible, authors should add the IH figures/photos of mast cells in TNBC tissues.

Author Response

Comment 1: The current study addressed the role, expression, and mechanisms/ biological effects of ANXA1 in triple negative breast cancers (TNBC). The paper is well written and comprehensive. There are only few minor points to address.

Response 1: We would like to thank Reviewer #1 for taking the time and effort to review our work and provide constructive criticism, which has significantly improved our manuscript. We have revised the manuscript following the Reviewer’s suggestions as outlined below.

Comment 2: line 91-92: unclear sentence: “…enriched inflammation and angiogenesis related gene sets,”…Do you mean increased expression/activation of .. gene sets? It is necessary to re-construct this sentence and make it easier for understanding.

Response 2: We agree with the Reviewer #1 that the original sentence was somewhat difficult to understand. We have revised the sentence as below.

Given this background, we hypothesized that ANXA1 high tumors are involved in increased mast cell infiltration, increased inflammation and angiogenesis related gene expression, and finally, are associated with worse survival in TNBC.

Comment 3: Figure 1: please specify in more details what does it mean:” High and low expression are represented by the red and blue lines” as in the figure legend it is marked as pos and neg. Does neg means low? And pos means high expression of ANXA? Correct if wrong.

Response 3: We would like to apologize for making critical typing errors in the description and figure legend for Figure 1 and the associated confusion that it caused. We amended the manuscript including the description in the Results section and corrected the Figure 1 legend as outlined below.

2.1. TNBC patients with positive ANXA1 protein stain had significantly worse overall survival (OS).

We have previously demonstrated that TNBC cases with high ANXA1 protein expression were associated with aggressive features of breast cancer (10). The limitation of that study was that all of the patients were treated prior to 2005 with a small cohort (n=35). We updated the cohort by adding 13 cases from 2005-2011 and re–analyzed the whole cohort. ANXA1 protein expression in breast cancer tissues was calculated by using anti-ANXA1 antibody staining (Figure 1A and B). Within the tumors, the ANXA1 expression was mainly detected in the cytoplasm of cancer cells, stromal cells and myoepithelial cells. Among 48 TNBC patients, 17 cases had positive ANXA1 staining. Overall survival (OS) was significantly shorter in patients with ANXA1 stain positive tumors compared with ANXA1 stain negative tumors (Figure 1). ANXA1 positivity in cancer cells had no statistical relationship with the clinical characteristics of the patients (Supplementary Table S1).

Positive and negative ANXA1 protein staining results are represented by the red and blue lines, respectively.

Comment 4: The Fig 1 A and B shows only negative and positive staining. However, the authors wrote about low and high expression levels of ANXA1. It is necessary to present how the Low and High expressing tissues look like. The Immunohistochemistry should be extended in fig 1. Sub-titles should be improved. For instance: “ANXA1 high tumors were associated with activated mast cells…” should be “ANXA1 high tumors were associated with localization/presence of activated mast…etc”. The staining/immunohistochemistry for mast cells presence will add a value to this paper. If it is possible, authors should add the IH figures/photos of mast cells in TNBC tissues.

Response 4: We would like to once again apologize for making critical typing errors in the description and legend for Figure 1 and the associated confusion that it caused. As the Reviewer pointed out, Fig 1 A and B shows negative and positive staining, which reflects ANXA1 protein production. We have fixed the error in Response 3. The other remaining data, including association with mast cell levels, were analyzed using ANXA1 gene expression data in either TCGA and/or METABRIC cohorts. Therefore, presence of mast cells was not identified by staining, but by calculation using a computational algorithm on transcriptomes. To this end, in addition to Response 3, we added the following sentence in the Discussion section as below.

In addition, immune cells were assessed with a computational algorithm alone and not with flow cytometry or IHC in this study.

Reviewer 2 Report

The article addresses a pertinent scientific question which carries along with it clinical implications. Introduction is simple and direct to the point. Despite, other sections require additional attention/modifications:

Regarding the section "Materials and Methods":

Patient cohorts

The sentence "high or low ANXA1 expression (...)confirmation of tumor recurrence" must be clarified. Why to choose the median of mRNA expression to be a definition of the cut-of? The concept of "disease-free survival" is not correct. Clinical samples for protein expression The reference of the study approval should be depicted in the text.

Regarding the section "Discussion":

The content of the paper should be subjected to further reflection granted by recent and significant literature on the field achieved by other authors and by some of the authors of this paper:

Mol Cancer Ther November 1 2017 (16) (11) 2528-2542 Leonardo A. Moraes, Patrick B. Ampomah & Lina H.K. Lim (2018) Annexin A1 in inflammation and breast cancer: a new axis in the tumor microenvironment, Cell Adhesion & Migration, 12:5, 417-423.

Regarding the section "Discussion":

This paragraph should be reformulated as the lack of causality between the high expression of ANXA1 and mast-cell induce inflammation and angiogenesis is stressed along the paper. Further conclusions cannot be pointed out as it is described once it is not clear the proper role of ANXA1 in the prior phenomena.

Author Response

Comment 1: The article addresses a pertinent scientific question which carries along with it clinical implications. Introduction is simple and direct to the point. Despite, other sections require additional attention/modifications:

Regarding the section "Materials and Methods":

Patient cohorts

The sentence "high or low ANXA1 expression (...)confirmation of tumor recurrence" must be clarified. Why to choose the median of mRNA expression to be a definition of the cut-of?

Response 1: We appreciate Reviewer 2’s comment that made us realize that our original manuscript lacked enough explanation. We chose the median because it is one of the most commonly used cut-off points used in previous reports that used the similar method. We added the references in the sentence.

Comment 2: The concept of "disease-free survival" is not correct.

Response 2: We agree with Reviewer 2 that our description in the original version was not accurate, and we have revised the description as outlined below.

Disease-free survival (DFS) was defined from the time of completion of primary treatment until clinical confirmation of tumor recurrence.

Comment 3: Clinical samples for protein expression

The reference of the study approval should be depicted in the text.

Response 3: We agree with Reviewer 2 that the study approval should be clearly depicted in the text. We added the following sentence to the Materials and Methods section.

This study was approved by the Ethics Committee of Fukushima Medical University.

Comment 4: Regarding the section "Discussion":

The content of the paper should be subjected to further reflection granted by recent and significant literature on the field achieved by other authors and by some of the authors of this paper:

Mol Cancer Ther November 1 2017 (16) (11) 2528-2542

Leonardo A. Moraes, Patrick B. Ampomah & Lina H.K. Lim (2018) Annexin A1 in inflammation and breast cancer: a new axis in the tumor microenvironment, Cell Adhesion & Migration, 12:5, 417-423.

Response 4: We appreciate Reviewer 2 for pointing out a recent and significant publication in the field that should be cited. We added the following sentences in the text. 

ANXA1 continues to attract attention, and some reported that the expression of ANXA1 may be associated with chemotherapy response in TNBC (Chen et al. Mol Cancer Ther 2017). Other authors reported that ANXA1 plays an important role in the tumor microenvironment (Moraes et al. Cell Adhesion & Migration 2018).

Comment 5: Regarding the section "Discussion":

This paragraph should be reformulated as the lack of causality between the high expression of ANXA1 and mast-cell induce inflammation and angiogenesis is stressed along the paper. Further conclusions cannot be pointed out as it is described once it is not clear the proper role of ANXA1 in the prior phenomena.

Response 5: We agree with Reviewer 2 that the second paragraph of the Discussion in our original manuscript sounded as if there is causality. We have revised our manuscript as seen below.

Strikingly, we found that ANXA1 high tumors significantly enriched angiogenesis gene sets where they associated with high infiltration of activated mast cells. The role of tumor infiltrating mast cells has been drawing attention in recent years, however, their impact on patients’ prognosis has been controversial (32, 33).

Reviewer 3 Report

In this manuscript Okano et al. attempt to determine the clinical relevance of ANXA1 in triple negative breast cancer by utilizing available large datasets using novel algorithms. They were able to show that high expression of this gene is associated with TNBC in comparison with other subtypes and demonstrates correlation with lower survival. The authors additionally mined the data using the novel immune cell sorting algorithm CIBERSORT in two cohorts (TCGA and METABRIC) that demonstrate association of mast cell infiltration with ANXA1 expression. GSEA analysis suggests that tumors with high ANXA1 expression enriched angiogenesis gene sets.

Major concerns

While the authors approached the validation of ANXA1 expression in the paper by two different methods such as RNA seq for initial screening and followed it up with immunohistochemistry such a two pronged approach was not utilized for the mast cell infiltration. Please clarify. The legend in Figure 1. Speaks of high and low expression of ANXA1 while the figure itself has pos and neg as the identifiers. Please clarify and keep consistent.

Author Response

Comment 1: In this manuscript Okano et al. attempt to determine the clinical relevance of ANXA1 in triple negative breast cancer by utilizing available large datasets using novel algorithms. They were able to show that high expression of this gene is associated with TNBC in comparison with other subtypes and demonstrates correlation with lower survival. The authors additionally mined the data using the novel immune cell sorting algorithm CIBERSORT in two cohorts (TCGA and METABRIC) that demonstrate association of mast cell infiltration with ANXA1 expression. GSEA analysis suggests that tumors with high ANXA1 expression enriched angiogenesis gene sets.

While the authors approached the validation of ANXA1 expression in the paper by two different methods such as RNA seq for initial screening and followed it up with immunohistochemistry such a two pronged approach was not utilized for the mast cell infiltration. Please clarify.

Response 1: We would like to thank Reviewer 3 for taking the time to review and for providing constructive comments. There was a technical reason why mast cell infiltration was not validated by IHC. The number of mast cells are so few that IHC analyses were unreliable. The gold standard assay is flow cytometry; however, no fresh samples were available for use. This limitation was added to the Discussion section as seen below.

It would have been ideal to validate the association of ANXA1 expression and mast cell infiltration with protein expression as it was previously done in survival analyses; however, no sample was available for flow cytometry that would allow such analyses.

Comment 2: The legend in Figure 1. Speaks of high and low expression of ANXA1 while the figure itself has pos and neg as the identifiers. Please clarify and keep consistent.

Response 2: We would like to apologize once more for making critical typing errors in the description and figure legend for Figure 1. Clearly this error has created major confusion as two of the Reviewers pointed it out. Our errors were corrected as described in Reviewer 1 - Response 3 above.
